

# Scale-adaptive simulation of wind turbines, and its verification with respect to wind tunnel measurements

Jiangang Wang, Chengyu Wang, Filippo Campagnolo, and Carlo L. Bottasso

Wind Energy Institute, Technische Universität München, Garching bei München, Germany

*Correspondence to:* C.L. Bottasso (carlo.bottasso@tum.de)

**Abstract.** This paper considers the application of a scale-adaptive simulation (SAS) CFD formulation for the modeling of single and waked wind turbines in flows of different turbulence intensities. The SAS approach is compared to a large-eddy simulation (LES) formulation, as well as to experimental measurements performed in a boundary layer wind tunnel with scaled wind turbine models.

The motivation for the use of SAS is its significantly reduced computational cost with respect to LES, made possible by the use of less dense grids. Results indicate that the two turbulence models yield in general results that are very similar, in terms of rotor-integral quantities and wake behavior. The matching is less satisfactory in very low turbulence inflows. Given that the computational cost is about one order of magnitude smaller, SAS is found to be an interesting alternative to LES for

repetitive runs where one can sacrifice a bit of accuracy for a reduced computational burden.

## 1 Introduction

Wake simulations provide for valuable and quantitative insight into the complex physics governing interactions among wind turbines and of wind turbines with the atmospheric boundary layer. A range of different models and numerical methods have been developed for wake interaction research,

spanning different fidelity levels.

At the lower end of this spectrum, engineering wake models are based on a few equations, parameterized in terms of a small number of tuning parameters. Jensen (1983) proposed a parametric wake deficit model based on momentum balance, while Jiménez et al. (2010) applied momentum conservation to establish a correlation between yaw misalignment and downstream wake skew an-

gle. Following this approach, the FLORIS model (Gebraad et al., 2016) combined expansion, wake skew and overlapping into a single engineering model. To improve the wake deficit formulation, Larsen employed Prandtl mixing length theory to derive a closed form solution (Renkema, 2007; Larsen et al., 2008). Later, Bastankhah and Porté-Agel (2016) derived a more sophisticated wake model based on the self-similar Gaussian profile to estimate the wake deficit for yaw misaligned

conditions. A dynamic wake meandering model (in contrast with the previous steady state models) was developed by Larsen et al. (2007) and validated by Churchfield et al. (2015). The formulation is





based on the solution of the unsteady axisymmetric thin-shear-layer equation to simulate the wake meandering effect.

At the higher fidelity end of the spectrum, large-eddy simulation (LES) models have been gaining an increasing level of attention, because of their foundation on first principles and their potential ability to more accurately model the physics at play (Calaf et al., 2010; Wu and Porté-Agel, 2011; Churchfield et al., 2012). These approaches usually include an atmospheric boundary layer generator for stable, neutral or convective conditions, a LES turbulence model of the flow, and a wind turbine model based on an actuator disk or actuator lines. Blade-conforming approaches are also being developed, where the rotor blades are represented not by their effects (through an actuator disk or line), but directly as rigid or flexible three-dimensional bodies interacting with the flow (Schulz et al., 2014). Initial verification attempts with field data have been presented in Wu and Porté-Agel (2015); Mirocha et al. (2015); Vollmer et al. (2015).

A much more extensive verification and validation activity of wake models is currently underway using scaled wind turbines in boundary layer wind tunnels. For example, Porté-Agel et al. (2011) used high-quality PIV measurement data to validate their LES approach.

Wang et al. (2018c)] have developed a LES framework based on SOWFA (Churchfield and Lee, 2012; Fleming et al., 2013), which has been used for simulating single and interacting scaled wind turbines operated in a boundary layer wind tunnel (Wang et al., 2018d, a). This LES approach has been systematically verified with respect to power, loads and wake measurements obtained in various wind tunnel experiments (Wang et al., 2017a, b, 2018c, a). However, the computational burden is significantly high and somewhat hinders its applicability to cases with many wind turbines, long physical times, or when multiple operating conditions are of interest, which is for example the case in wind plant control research. In fact, dense meshes are typically necessary for resolving wakes and their interactions with the downstream machines.

In an attempt to address the bottleneck caused by the computational effort of a LES approach, the present paper studies the applicability of the scale-adaptive simulation (SAS) approach as an alternative —faster— turbulence model for wake interaction simulation. SAS (Menter and Egorov, 2010; Egorov et al., 2010) is a turbulence model derived from Menter's $k$-$\omega$ shear stress transport (SST) model, which consists of an additional production term in the $\omega$ transport equation. This enables the specific dissipation rate $\omega$ to adapt to local flow inhomogeneities through the von Kármán length scale $L_{vK}$. The SAS model applied to homogenous turbulence conditions results in a typical SST behavior. On the other hand, in the case of non-homogeneous turbulence —as in wind turbine wakes— the von Kármán length does introduce the effects of the second velocity derivative into the eddy viscosity formulation. This way, the SAS model exhibits a behavior similar to LES, but at a cost comparable to URANS (unsteady Reynolds-averaged Navier-Stokes).

Wang et al. (2018b) presented an initial application of SAS to one single and two aligned wind turbines in a low-turbulence inflow environment. Some encouraging results were obtained, indicat-





ing that SAS is indeed capable of properly estimating velocity and turbulence intensity profiles of
turbine wakes at a much reduced computational cost, although blade tip vortices would have required
denser grids to be fully resolved. Goal of this paper is to extend that work, by more systematically
verifying the SAS model for wake interaction simulation, including higher turbulence flows, yaw
misalignment conditions, and multi-wind turbine configurations.

The paper is organized according to the following plan. The SAS model is introduced in section 2,
the computational setup is reported in section 3 —including turbulent inflow generation and wind
turbine simulation—, while the experimental setup is presented in section 4. Results are presented
and discussed in section 5, and include a single wind turbine aligned (§5.1) and misaligned (§5.2)
with respect to a low turbulence inflow, and in moderate and high turbulence conditions (§5.3). A
case consisting of three aligned wind turbines is used to assess the method in wake overlapping
conditions (§5.4). Finally, conclusions are drawn in section 6.

## 2 Numerical model

### 2.1 CFD formulation

The numerical model is developed based on `SOWFA` (Churchfield and Lee, 2012), a simulation tool
derived from the standard Boussinesq PISO (Pressure Implicit with Splitting of Operator) incom-
pressible solver in the OpenFOAM repository. A blended spatial differencing algorithm, coupled
with the Gamma (Jasak et al., 1999) and central differencing scheme is implemented in the solver
to limit numerical dispersion as well as numerical diffusion; the Gamma scheme is used in the near
wake, while central differencing in the far wake. The second order implicit backward scheme is used
for temporal discretization. The pressure ($p$) equations are solved by a conjugate gradient, precon-
ditioned by a geometric-algebraic multi-grid, while a bi-conjugate gradient is used for the resolved
velocity $\bar{\mathbf{u}}$, dissipation rate $\omega$ and turbulent kinetic energy $k$, using the diagonal incomplete-LU fac-
torization as preconditioner.

The framework uses the actuator-line method (ALM) to represent the effects of the blades, imple-
mented according to the velocity sampling method of Churchfield et al. (2017). The implementation
of the actuator lines is obtained by coupling the CFD solver with the aeroservoelastic simulator
`FAST` (Jonkman and Buhl Jr, 2005).

An immersed boundary (IB) formulation (Wang et al., 2017b) is employed to model the wind
turbine nacelle and tower. Compared to an actuator line model of these two components, IB signif-
icantly improves the near wake performance and the accuracy of higher order quantities (vorticity
and turbulence intensity) (Wang et al., 2017b).



### 2.2 SAS model

The derivation of the SAS model follows the work of Menter and Egorov (2005, 2006); Egorov and Menter (2008) and Lindblad et al. (2014), and it is formulated in incompressible form with minor modifications.

Similar to the $k$-$\omega$ SST model, the kinematic eddy viscosity is modeled as $\nu_t = k/\omega$. In the SAS model, however, an additional source term $Q_{\text{SAS}}$ is introduced to improve $\nu_t$. The $k$ and $\omega$ transport equations of the SAS model read

$$\frac{\partial k}{\partial t} + \nabla \cdot (\tilde{\mathbf{u}} k) = P_k - c_\mu k \omega + \nabla \cdot \left( \left( \nu + \frac{\nu_t}{\sigma_k} \right) \nabla k \right), \tag{1a}$$

$$\frac{\partial \omega}{\partial t} + \nabla \cdot (\tilde{\mathbf{u}} \omega) = \alpha \frac{\omega}{\rho k} P_k - \beta \omega^2 + Q_{\text{SAS}}$$

$$+ \nabla \cdot \left( \left( \nu + \frac{\nu_t}{\sigma_\omega} \right) \nabla \omega \right) + (1 - F_1) \frac{2}{\sigma_{\omega_2}} \frac{1}{\omega} \nabla k \cdot \nabla \omega, \tag{1b}$$

where $c_\mu$ and $\sigma_k$ are the closure coefficients in the $k$ equation, while $\alpha$, $\beta$, $\sigma_\omega$ and $\sigma_{\omega_2}$ are the closure coefficients in the $\omega$ equation. $F_1$ is a blending function that transitions between the $k$-$\omega$ and $k$-$\epsilon$ modes. The dissipation terms $c_\mu k \omega$ and $\beta \omega^2$ are discretized in implicit form to improve convergence and stability (Lindblad et al., 2014). The present implementation does not consider any prediction-correction iteration between the momentum and the two transport equations: in the $k$ and $\omega$ equations, $\tilde{\mathbf{u}}$ is considered as a known velocity field resolved by the PISO scheme, solving the transport equations in segregated form (Lindblad et al., 2014). Therefore, the implementation of SAS does not require a change in the PISO algorithm.

Through the boundary layer length scale, the blending term is in charge of shifting between boundary layer and free-stream type conditions, which is the main idea behind the SST approach. Such formulation, however, cannot detect local flow inhomogeneities and requires the modeling of an additional source term. The $Q_{\text{SAS}}$ term originates from Rotta's $k$-$kL$ model of the correlation-based length scale (Egorov and Menter, 2008) , and it is formulated as

$$Q_{\text{SAS}} = F_{\text{SAS}} \cdot \max \left( \zeta_2 \kappa S^2 \left( \frac{L}{L_{vK}} \right)^n - C \frac{k}{\sigma_\Phi} \max \left( \frac{|\nabla \omega|^2}{\omega^2}, \frac{|\nabla k|^2}{k^2} \right), 0 \right), \tag{2}$$

where parameters $\zeta_2$, $\sigma_\Phi$ and $C$ were obtained from experiments. $F_{\text{SAS}}$ behaves as a scaling parameter that dictates the amount of numerical damping injected in the flowfield. The value of $F_{\text{SAS}}$ requires specific calibration. $L$ is the length scale of the modeled turbulence, and $L_{vK}$ the von Kármán length scale. In order to preserve the SST characteristics of the formulation, the $Q_{SAS}$ term is defined as a strictly positive term. Regarding the scaling ratio $(L/L_{vK})^n$, numerical experiments have shown that the linear length scale ratio (also used in Egorov and Menter (2008)) leads to better stability and robustness of the formulation when compared to the quadratic form of Menter and Egorov (2010); Egorov et al. (2010). The choice $n = 1$ is therefore adopted in this work.



The von Kármán length scale $L_{vK}$ is formulated based on Rotta's equation, and it writes

$$L_{vK} = \max\left( \frac{\kappa S}{|\nabla^2 \tilde{\mathbf{u}}|}, C_k \Omega_{CV}^{1/3} \right), \tag{3}$$

ensuring the scale-adaptive characteristics of the method (Menter and Egorov, 2005). In fact, $L_{vK}$ reflects the size of resolved eddies in the flow, while the SST model only considers length-scales associated with the boundary layer thickness.

The second derivative of the velocity field $\nabla^2 \tilde{\mathbf{u}}$ detects inhomogeneities in the resolved turbulence scales, and brings this information into the eddy viscosity term. As mentioned earlier, the SST model

is only associated with the boundary layer length scale (i.e., the $F_1$ term), but it is not adjusted to the local flow characteristics (Menter and Egorov, 2005; Younsi et al., 2008). Therefore, when the SST model is used in free-stream high-Reynolds wake flows, it cannot adapt $\omega$ in the wake regime, often resulting in a highly diffusive behavior. By introducing $\nabla^2 \tilde{\mathbf{u}}$, the improved formulation is capable of adjusting the eddy viscosity to better control the amount of numerical diffusion to the

local characteristics of the flow. In addition, to avoid an insufficient damping at high wave numbers, a lower limit to $L_{vK}$ is imposed, which is proportional to the model parameter $C_k$ and the cell size $\Omega_{CV}^{1/3}$, $\Omega_{CV}$ being the cell volume.

In summary, the formulation of $L_{vK}$ achieves a balance between the production and destruction of turbulence kinetic energy, providing for a suitable numerical diffusion at the sub-grid scales.

By introducing the $Q_{\text{SAS}}$ term into the $\omega$ equation, the SAS model improves the SST formulation on three aspects: an improved modeling of the eddy viscosity, a more accurate prediction of the breakdown of turbulent structures, and a better high wave number damping for the resolved eddies down to the grid limit (Egorov and Menter, 2008). Such improvements eventually lead to a LES-like behavior of the SAS formulation.

## 3  Computational setup

The simulation model represents a complete digital copy of experiments conducted in a boundary layer wind tunnel, including the passive generation of a sheared and turbulent flow and its interaction with scaled wind turbines. Since a same turbulent flow can be used for different turbine simulations, the computational domain is subdivided into two partitions: a precursor simulation, charged with

the modeling of the flow evolution along the tunnel, and a successor simulation —whose inflow is obtained from the precursor— modeling the wind turbines.

### 3.1  Precursor simulation

The precursor simulation is used to generate the turbulent inflow for a successive wind turbine run. The precursor domain uses a structured body-conforming volume mesh (entirely composed of hex-

ahedral elements) to discretize the volume around turbulence-generating spires at the wind tunnel





inlet, as well as the rest of the wind tunnel test section. Only the LES model is employed for precursor simulations, as generating a turbulent inflow is not a repetitive task. Therefore the reduction of its computational costs is not a priority, while the resulting flow should be of the highest possible quality.

The mesh density is designed not to fully resolve the boundary layer, and the average $y^+$ is equal to 50. In fact, a wall-modeled simulation significantly reduces the computational costs compared with a fully resolved one. The precursor mesh contains 59 million cells with the current setup, and it would require one or two orders of magnitude more cells for $y^+$ close to 1. As shown later on, this approach is still capable of a good matching with experimental measurements.

There are two types of turbulence-generating spires: one is used to generate a moderate turbulence ($\approx 6\%$), while the second type is for high turbulence ($\approx 20\%$) conditions. The time-averaged wind speed at hub-height is equal to 5 m/s for both cases. The maximum Courant number is limited to one. After reaching steady state conditions, which takes about 15 s of physical time, the flow field is recorded at a plane 19 m downstream of the tunnel inlet, to serve as input for subsequent wind

turbine simulations. A detailed description of the precursor simulation setup is given in Wang et al. (2018c).

In this paper, also a low turbulence inflow condition is considered. In that case, the inflow to the wind turbine simulation is not obtained by a precursor, but simply by prescribing a time-constant velocity measured by scanning LiDARs 29 m downstream of the tunnel inlet (Van Dooren et al.,

180   2016).

### 3.2  Wind turbine simulation

#### 3.2.1  Computational mesh

The computational setup for the wind turbine simulation follows Wang et al. (2018b). The domain layout is shown in Fig. 1, which is used for both the low (standalone) and moderate/high (coupled

with the precursor inflow) turbulent conditions. The domain width is reduced to 3.6D, which is 4.2 times shorter than the test section width to reduce the computational cost while avoiding blockage effects. A coordinate reference frame is centered at the hub of the front turbine, as shown in the same figure. The mesh uses three different densities: zone 1 is the base mesh, with cubic cells of 0.08 m in size, while the cells in zone 2 have a size of 0.04 m. Zone 3 is the finest grid, used in close proximity

of the wind turbines and their wakes. The cell size in this region of the domain differs depending on the turbulence model: for LES the cells have a size of 0.01 m, while for SST and SAS the size is 0.02 m. As a result, the SST and SAS models are run on a grid that has about 7.8 times fewer cells than in the LES case (which has a total cell number of about 39 millions). In all cases, cells are cubic except for polihedral elements used for connecting together the zone boundaries, which

however account for less than 1% of the total cell count.





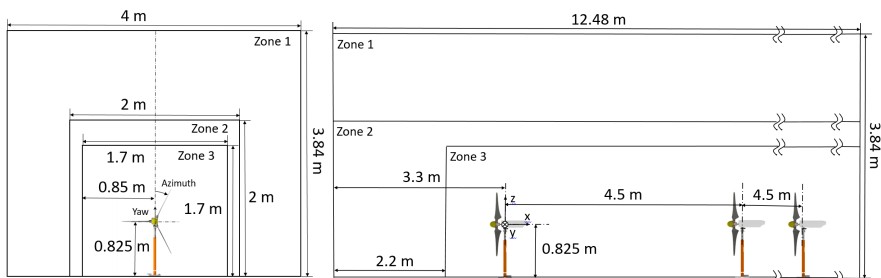

**Figure 1.** Experimental layout and computational domain.

As shown later on, results indicate that the velocity and turbulence intensity fields are very similar between LES and SAS in moderate and high turbulence conditions. In this sense, SAS is capable of achieving a performance similar to LES, but with much coarser grids. As a verification of mesh convergence, the SAS model was also run on the finer LES grid, obtaining essentially identical

results to the ones found on its coarser grid.

On the other hand, finer grids would be necessary for LES for a more accurate solution of the low turbulence case. As discussed later in this work, low turbulent conditions have much more stringent requirements in the resolution of the near wake vortical structures and their breakdown. Such a high accuracy is not needed at higher turbulence, as this becomes the dominating factor that dictates

vortex breakdown. Finer grids were however not used here, because of the dramatic increase in computational cost caused by the current structured grid approach.

For both SAS and LES, the grid implicitly operates a spatial filtering on the solution. Similarly, although a temporal filtering is implicitly performed by the time marching algorithm, no explicit temporal filtering is applied a priori on the solution, in contrast to the URANS model. In this regard,

SAS can be considered as a LES dynamic sub-grid scale model, which exhibits a LES-like behavior at the resolved scales (Menter and Egorov, 2005).

### 3.2.2 Boundary conditions

The same boundary conditions for the flow velocity $\tilde{\mathbf{u}}$, pressure $p$ and temperature $T$ are used for the SST, SAS and LES methods. The domain inlet uses the velocity inflow map obtained from either

the LiDAR scanned inflow map (Van Dooren et al., 2016) (low turbulence case) or the turbulent precursor simulation (moderate and high turbulence cases). Boundary conditions on the left/right side walls are as follows. Since the domain width is reduced with respect to the actual one, Neumann type surface conditions are set for $p$ and $T$, while mixed type wall conditions are used for $\tilde{\mathbf{u}}$, setting the wall-normal velocity component to zero to ensure mass conservation and numerical stability.

The ceiling and floor use Dirichlet-type non-slip wall conditions. The IBs of nacelle and tower use



Dirichlet-type non-slip wall conditions for the low turbulence condition, while slip wall conditions are used in the turbulent cases on account of numerical stability issues.

Apart from the resolved flow quantities, boundary conditions are also set for sub-grid scale quantities. The constant Smagorinsky LES model uses Neumann and Dirichlet-type surface conditions for eddy viscosity $\mu_t$ at left/right and ceiling/floor faces, respectively. In the Dirichlet case, a value equal to $1 \times 10^{-5}$ m$^2$/s is used at the centroids of boundary cells to account for the negligible turbulence near the surface. The eddy viscosity $\mu_t$, on the other hand, is the ratio of the two additional variables $k$ and $\omega$ for the SST and SAS models. Dirichlet-type wall conditions are imposed on both the ceiling and floor surfaces for $k$ and $\omega$, using the values $1 \times 10^{-4}$ and $2 \times 10^{-2}$, respectively, based on Menter and Esch (2001) and simulation stability tests. Results are also largely insensitive to the boundary values for $k$ and $\omega$, which therefore do not require a precise calibration.

### 3.2.3 Numerical implementation

The SAS and SST models implement the same linear solvers used in the LES case for the resolved scales, as described in §2.1. Regarding sub-grid scale quantities, central differencing could in principle be used for the convective terms of the $k$ and $\omega$ equations for both the SAS and LES models. However, due to the von Kármán length scale in the $\omega$ equation of the SAS model, oscillations may be generated that, by affecting the eddy viscosity, can eventually cause the simulation to diverge. Accordingly, a strictly bounded Van Leer differencing scheme is used for the $k$ and $\omega$ transport equations to minimize numerical stability issues. Although such a discretization may cause significant numerical diffusion, boundedness should be favored over accuracy for a scalar field scheme (Greenshields, 2015). The $k$ and $\omega$ equations for both the SST and SAS model are solved by the conjugate gradient algorithm with diagonal incomplete-LU preconditioning.

### 3.2.4 Parameter tuning

The time step length for all three turbulence models is limited by imposing that the lifting line representing the blade does not cross more than one cell in one step (Martinez et al., 2012), which is equivalent to 0.3 of the maximum Courant number. Since the size of the smallest cells for the LES grid is half that of the SAS and SST ones, the time step is accordingly smaller. The difference in grid size (and hence the number of cells) and in time step length are the main factors driving the different computational costs of these methods. In fact, although SAS and SST require the solution of two additional transport equations, the resulting additional cost is negligible when compared to the effects of grid size and time step. On average, SST and SAS are roughly 13 times faster than LES for the simulation cases considered here.

The Gaussian width $\epsilon$ of the ALM is set to 2.5 times the cell size at the rotor disk, which once again results in different values on account of the different grid densities of the turbulence models. The constant Smagorinsky model is employed for the LES case, using $C_s$ equal to 0.13. The scal-





ing parameter $F_{\text{SAS}}$ is set to 2 for the SAS model. Tests have shown that the performance of the SAS model is dependent on this parameter, so that its careful calibration becomes essential for the accuracy of the results.

## 4   Experimental setup

Experiments were conducted in a 36 m by 16.7 m by 3.84 m boundary layer wind tunnel at Politecnico di Milano (Bottasso et al., 2014). The scaled wind turbine are of the G1 model type, with a rotor diameter of 1.1 m and more completely described in Campagnolo et al. (2016c, a, b, 2018)]. Table 1 illustrates the main characteristics of the machine. The hub-height wind speed for the three operating conditions (low, moderate and high turbulence inflow) is lower than the rated speed of the

wind turbine, so that blades were set to their fine pitch setting. The turbulence intensity is 2% for the low turbulence condition, while it is 6% and 20% at hub height for the moderate and high turbulence ones, respectively.

A look-up table torque controller in the loop was used in the experiments, while a fixed rotating speed equal to the average experimentally measured one was used in the simulations. In reality, the

angular speed of the rotor will vary because of turbulent fluctuations in the flow field. It was verified that, as expected according to intuition, imposing a constant rotor speed in a CFD-ALM simulation does significantly affect the estimation of loads (Wang et al., 2018a). However, it was also verified that such simplification does not significantly influence the downstream wake, which is the focus of this paper.

Triple hot-wire anemometers were used to measure the flow velocity components at several distances behind the rotor. Sensors integrated onboard the scaled wind turbine were used to measure the instantaneous rotor torque, tower base bending and rotating speed.

**Table 1.** Main characteristics of G1-type wind turbine model.

| Type | G1 |
| --- | --- |
| Rotor orientation | Upwind |
| Rotor diameter | 1.1 m |
| Rated speed | 6.06 $\text{ms}^{-1}$ |
| Rated angular speed | 850 RPM |
| Rotating direction | Clockwise, facing downstream |
| Control | Variable speed, torque, pitch and yaw control |





## 5 Result and analysis

Three quantities are used to evaluate the simulation accuracy with respect to experimental measurements. The average velocity $\langle u \rangle$ is computed based on time-averaged velocity components sampled along a hub-height horizontal line in the flowfield, which are then spatially averaged along the rotor diameter. The percentage average velocity error is defined as $\langle \Delta(u) \rangle = (\langle u_{\text{sim}} \rangle - \langle u_{\text{exp}} \rangle)/\langle u_{\text{exp}} \rangle$. The root mean square (RMS) error is used to quantify the spatial fit between simulations and experiments (Chai and Draxler, 2014), and it is defined as

$$\text{RMS}(u) = \sqrt{\frac{1}{N} \sum_{j=1}^{N} \left( \left\langle u_{\text{sim}}^{j} \right\rangle - \left\langle u_{\text{exp}}^{j} \right\rangle \right)^{2}}, \tag{4}$$

where $\langle u^j \rangle$ is a time-averaged velocity component at a given spatial point $j$. The calculation of turbulence intensity $\sigma/\langle u \rangle$ needs to account for the turbulence both at the resolved and modeled scales. To this end, modeled fluctuations are summed to the resolved ones, yielding

$$\frac{\sigma^j}{\langle u^j \rangle} = \frac{\sqrt{\frac{1}{N} \sum_{i=1}^{N} (u^{i,j} - \langle u^j \rangle)^2 + \langle 2/3\,k^j \rangle}}{\langle u^j \rangle}, \tag{5}$$

where $u^{i,j}$ represents a velocity component at a given spatial point $j$ and at time step $i$. The term $\sqrt{\langle 2/3\,k^j \rangle}$ is the velocity fluctuation corresponding to the modeled turbulence kinetic energy $k^j$. In addition to the point-wise turbulence intensity $\sigma^j/\langle u^j \rangle$, rotor-average turbulence intensity $\sigma/\langle u \rangle$ and turbulence intensity $\text{RMS}(\sigma/\langle u \rangle)$ are defined similarly to the velocity case.

### 5.1 Single-turbine baseline case

A first baseline case is used to tune the parameters for the three turbulence models, parameters that are then used unchanged in the other cases considered herein. The baseline case represents an isolated flow-aligned wind turbine operating in a low turbulence environment in the partial load region. The CPU time ratio of LES and SAS for this case is $\text{CPU}_{\text{LES}}/\text{CPU}_{\text{SAS}}$=16.9.

The wind turbine power measured in the experiment is 45.8 W, while for SST, SAS and LES it is 44.8 W, 45.1 W and 45.5 W, respectively. Hence, the power output predicted by SAS appears to be in good agreement with both LES and measurements. As previously stated, the same ALM Gaussian width in terms of cell size is used for all methods.

Figure 2 shows vorticity contours for the LES, SAS and SST simulations, respectively from left to right. It appears that LES is capable of a significantly higher resolution of the tip and root vortices than SAS, thanks to its denser grid. Additionally, a much higher vorticity is produced by SAS compared to SST, due to its enhanced ability of resolving small scale features.

For a more precise understanding of these different representations of vorticity and of the overall modeling of wake structures, experimental measurements at hub height and at different distances downstream of the rotor are considered. Figure 3 shows the normalized time-averaged longitudinal




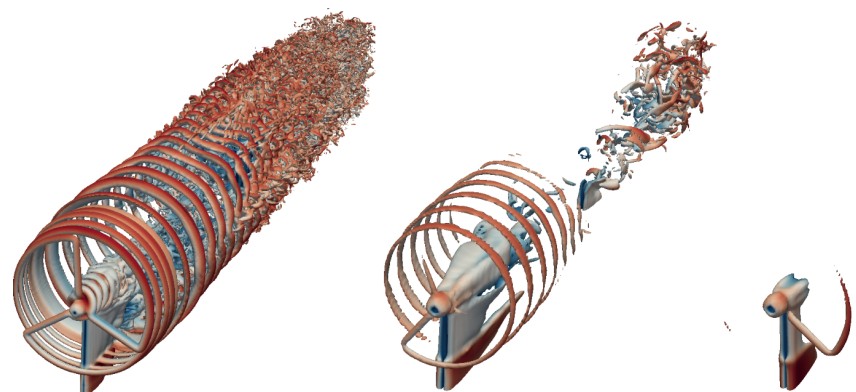

**Figure 2.** Vorticity iso-surface plots ($|\nabla \times \mathbf{u}| = 40$ 1/s) for LES, SAS and SST (from left to right).

velocity (top) and turbulence intensity (bottom) profiles at several downstream distances for LES,
SST, SAS and experiment.

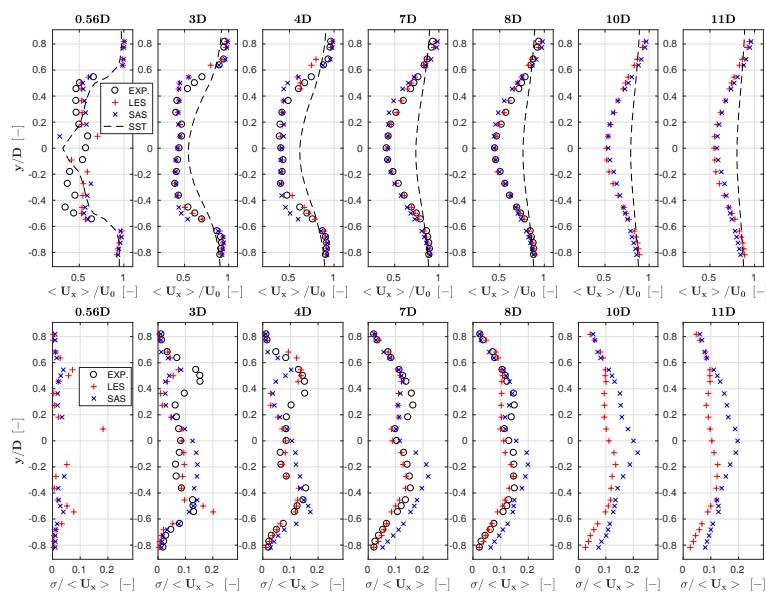

**Figure 3.** Normalized time-averaged stream-wise velocity (top) and turbulence intensity (bottom) profiles at
hub height and at several downstream positions. Experiment: black ○ symbols; LES: red + symbols; SAS: blue
× symbols; SST: black dashed line.

The average percent error $\langle \Delta(u_x) \rangle$ between simulation and experiment for LES is equal to -2.7%,
-1.6% and -1.1% at 3D, 4D and 8D, respectively. For SAS, the error is -4.1%, -5.4% and -3.3% at





the same downstream positions. RMS($u_x$) for LES equals 0.34 m/s, 0.33 m/s and 0.12 m/s at 3D,

4D and 8D, respectively, and it is 0.44 m/s, 0.50 m/s and 0.25 m/s for SAS at those same locations. The SAS velocity profiles show a reasonable agreement with both LES and the experimental curves.

On the other hand, the error at the same distances for SST is 11%, 13% and 13%, which is significantly larger than for SAS. The SST RMS($u_x$) is also on average twice as large than for SAS. These results suggest that, by including local flow inhomogeneities through $\left|\nabla^2\tilde{\mathbf{u}}\right|$, the modeling

of the wake is significantly improved, by a locally adjusted eddy viscosity and limited numerical diffusion.

It is interesting to observe that, close to the rotor (0.56D), SST and SAS predict nearly the same speed profile. In fact, in the near wake region the flow is not yet strongly affected by mixing and numerical diffusion, so that differences in the modeling of unresolved scales play a lesser role. In

this region of the wake, the behavior is mostly governed by the rotor thrust, which indeed is quite similar for all three models. The 10 s time-averaged thrust is in fact 16.1 N, 15.9 N and 15.7 N for LES, SAS and SST, respectively. On the other hand, moving downstream away from the rotor, the overestimated eddy viscosity of the SST model begins to show its effects on the wake deficit, as apparent in the plots starting at the 3D location all the way to the end of the domain.

It should be noticed that at 3D and 4D the velocity profiles of LES match very well those of the experiments, while SAS predicts a slightly larger wake width. This phenomenon is due to a lack of resolution of the blade tip vortices. Further downstream, the tip vortices collapse and break down, and therefore this effect is reduced. In particular, SAS curves show a very good match with LES at 10D and 11D. This indicates that numerical diffusion is well controlled by the SAS model

throughout the propagation of the wake, and flow mixture is properly resolved.

It should also be remarked that the resolution of tip vortices plays a lesser role than in the present case for moderate/high turbulence inflows. In fact, in those conditions vortex breakdown will take place earlier due to the higher background ambient turbulence. Therefore, the accuracy of SAS is expected to improve for more turbulent cases, as in fact confirmed by results shown later on in

this work. From this point of view, this initial baseline scenario represents a particularly difficult problem.

LES underestimates turbulence intensity by 23% and 12% at 3D and 4D, respectively, while for SAS the error is 11% and 10%. The consistent underprediction of turbulence intensity for low turbulence inflow conditions in the near wake region has already been observed by Troldborg et al.

(2015). However, results are quite similar for the two models considered here, which indicates the ability of SAS in resolving second order quantities in the near wake region.

There is a significant lack of symmetry in the profiles left (looking downstream, i.e. for positive $y$ values) and right of the rotor axis for both methods, as in fact the left peak is significantly underpredicted. This lack of symmetry however does not appear in the experimental results. This is probably

due to a combination of lack of resolution of the tip vortices and their interaction with the wake shed




by nacelle and tower. This problem is analyzed in detail further on, in reference to a yaw misaligned case. The effects of this lack of symmetry on turbulence intensity is consistent with a small lack of symmetry in wake recovery. In fact, especially at 7D and 8D, the numerical velocity profiles exhibit a reduced wake recovery on the left of the wake compared with the experimental measurements.

This fact is attributable to the lower upstream turbulence intensity on this same side of the wake, shown in the bottom row of plots of Fig. 3.

Regarding the far wake at 10D and 11D, SAS overestimates turbulence intensity by more than 50%, which may lead to a faster wake recovery further downstream. However, such a problem is only limited to low turbulence conditions, and the situation improves for higher turbulence.

Since SST is clearly unable to provide for sufficiently accurate estimates of the wake behavior, it is not considered further in the present work.

### 5.2 Single-turbine yaw-misaligned case

A correct estimation of wake behavior in yaw misaligned conditions is crucial, especially when significant intentional misalignments are generated for wake deflection wind plant control. LES and

SAS are compared to experiments, by considering hub-height velocity profiles 4D downstream of the rotor for six different yaw misalignment angles, namely $\pm 5$ deg, $\pm 10$ deg and $\pm 20$ deg. The flow conditions are the same low turbulence ones of the baseline case. Therefore, as previously observed, SAS results are somewhat affected by a lack of resolution of the tip vortices in the near wake region. The CPU time consumption ratio for this case is $\text{CPU}_{\text{LES}}/\text{CPU}_{\text{SAS}}$=14.4.

Figure 4 shows a comparison of the velocity (top) and turbulence intensity (bottom) profiles for the considered yaw misalignment angles. The velocity profile of the SAS model shows a relatively good agreement with both the experiment and LES. The maximum average-velocity error for SAS is 5.2% at -10 deg, while it is 4.1% for LES at 20 deg. The overall error over the six yaw configurations is 4% and 1% for SAS and LES, respectively. Likewise, the maximum $\text{RMS}(u_x)$ for SAS is 0.54 m/s

and it is 0.35 m/s for LES. The overall $\text{RMS}(u_x)$ over the 6 yaw configurations is 0.45 m/s and 0.29 m/s for SAS and LES, respectively. As noted in the baseline case, SAS overpredicts the wake width, due to a lack of resolution of the tip vortices. However, other than this, it is in a reasonably good agreement with LES.

The overall average turbulence intensity error over the 6 yaw configurations is 11% and 20%

for SAS and LES, respectively. In both cases, turbulence intensity is not everywhere matching well with the experiments. Here again, one of the two external peaks of the profiles is typically severely underpredicted by both methods, similarly to what was observed for the baseline case.

In fact, the turbulence intensity peaks correspond to the blade tip region, where the mesh is not fine enough for an accurate modeling of the tip vortices. For LES, the blade tip chord length is 1.8

times the cell size, which is clearly not enough to precisely resolve the tip vortices. The situation would be clearly much different for a body conforming blade meshing approach, however at the cost



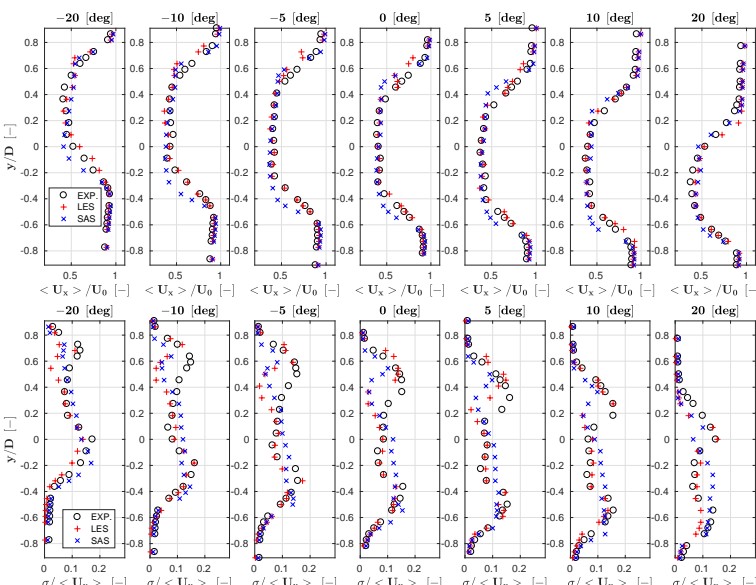

**Figure 4.** Normalized time-averaged stream-wise velocity (top) and turbulence intensity (bottom) profiles at hub height and 4D downstream, for various yaw misalignment angles. Experiment: black ○ symbols; LES: red + symbols; SAS: blue × symbols.

of an increased computational effort. The left peak in Fig. 4 is particularly much lower than in the experimental case. Since the SAS cell size at the blade tip is twice as large as in the LES case, this peak for SAS is even lower than the one for LES. The same phenomenon can also be observed in Fig. 5, where the blade tip vorticity and turbulence intensity contours all indicate that the SAS model is less capable of resolving tip vortices than LES.

On the other hand, the turbulence intensity peak on the right is well predicted by both SAS and LES. This phenomenon can be explained by the interaction of the wake shed by the tower with the tip vortices. Since the rotor spins in a clockwise direction (looking downstream), the wake has a counterclockwise swirling motion, as shown by the two bottom plots of Fig. 5. In turn, this causes the tower wake to move slightly to the right and upwards, increasing the turbulence intensity at hub height in this region of the wake. This effect is well illustrated by the LES turbulence intensity color plot of Fig. 5. The higher turbulence intensity on the right of the wake promotes a faster decay of the tip vortices than on the left, as shown by the vorticity color plot of the same figure. Because of this enhanced mixing, it is not necessary to have an extremely fine resolution of the grid, and even the relatively coarse mesh used here is enough to capture reasonably well the turbulence intensity peak on the right of the wake. The situation is different on the left: here, there is a very low background turbulence, as the incoming flow is almost uniform and there is little or no effect from the tower




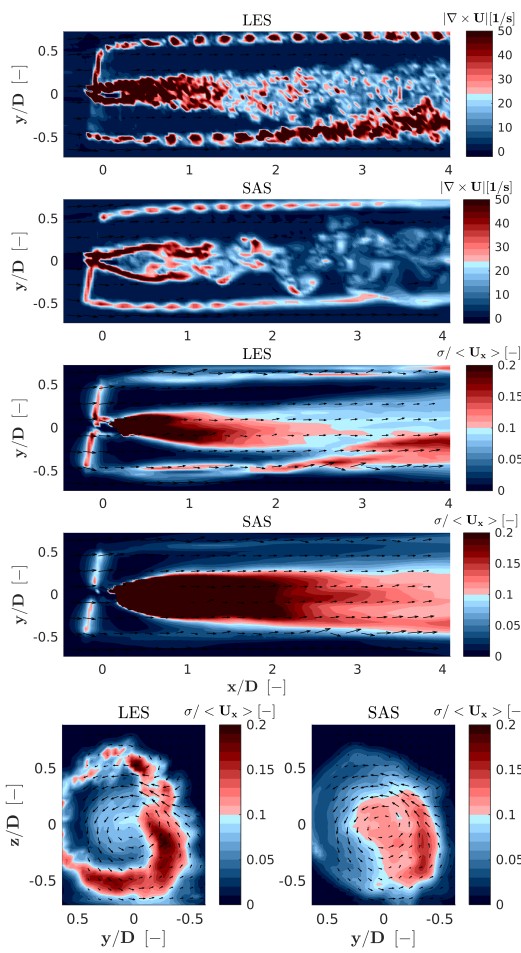

**Figure 5.** From top to bottom, vorticity magnitude $|\nabla \times \mathbf{u}|$ and turbulence intensity $\sigma/\langle u_x \rangle$ on a hub-height horizontal plane, and turbulence intensity on a vertical plane (looking downstream) 4D behind of the rotor. Black arrows indicate the cross-flow velocity vector at a number of sampling points.



wake. Hence, to estimate the correct turbulence intensity one would have to resolve very accurately
the tip vortices, something that is however not possible with the current grid density.

This interpretation of the results was confirmed by a simulation conducted without nacelle and
tower. In that case, which is not reported here for brevity, very similar turbulence intensity peaks
were observed to both the right and left of the wake.

The good matching of the right peak deserves a further comment. In fact, here the turbulence
intensity matches well the experiment, while the results on the left peak demonstrate a general lack
of resolution of the tip vortices. Hence, to compensate for this, the turbulence generated by the
immersed boundaries of tower (and nacelle) must probably be overestimated. Indeed, this is very
probable, based on the large difference between SAS and LES (which has a twice as dense mesh) in
turbulence intensity in the hub-height core of the wake shown in Fig. 5.

The analysis can be applied to the baseline case described in §5.1, which shows a very similar
behavior of the turbulence intensity peaks.

### 5.3   Turbulent cases

#### 5.3.1   Single-turbine moderate-turbulence case

Next, an isolated wind turbine is considered in a moderate turbulence environment. The turbulent
inflow is generated by the precursor simulation, as described in §3.1. The machine is aligned with
the flow and operates at the fixed rotating speed of 720 RPM with a collective pitch of 1.4 deg.
The numerical models use the same exact parameters employed for the low turbulence cases. The
CPU time ratio between the two turbulence models is $\text{CPU}_{\text{LES}}/\text{CPU}_{\text{SAS}}$=9.37 in this case. The 60 s
average rotor power is equal to 31.0 W for the experiment, 30.5 W for LES and 30.1 W for SAS.

Figure 6 shows the normalized velocity and turbulence intensity profiles for the experiment, LES
and SAS at -1.5D, 1.4D, 1.7D, 2D, 3D, 4D, 6D and 9D. The first measurement station at -1.5D is
upstream of the wind turbine, where the flow can be regarded as the undisturbed free stream. The
velocity profiles are all, in general, in a good agreement with one another. The overall simulation
error $\langle \Delta(u_x) \rangle$, averaged over all distances, is equal to 0.9% for LES and it is 1.1% for SAS. At
4D downstream, where a second wind turbine is located in other experiments, $\langle \Delta(u_x) \rangle$ is 2.1% for
LES and 3.2% for SAS. Throughout the wake propagation from near (1.4D) to far wake (9D), the
$\text{RMS}(u_x)$ for LES gradually reduces from 0.18 m/s to 0.13 m/s, while for SAS it decreases from
0.21 m/s to 0.08 m/s. Comparing to the low turbulence case of §5.1, the RMS values are drastically
reduced, which indicates a significant increase of the simulation accuracy for the present moderate
turbulence condition.

From 1.4D to 9D, the average turbulence intensity error for SAS is 2.6%, 5.1%, 6.3% 10.7%,
12.1%, 8.2% and 7.6%, respectively. Contrary to the low turbulence (baseline) case, turbulence in-
tensity for SAS significantly improves in the far wake (10D and 11D) in the moderate turbulence





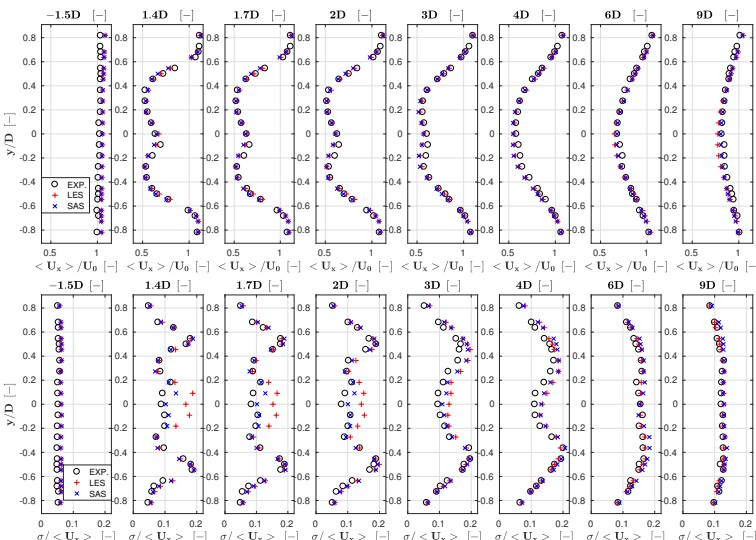

**Figure 6.** Normalized time-averaged stream-wise velocity (top) and turbulence intensity (bottom) profiles at hub height and at several positions. Experiment: black ○ symbols; LES: red + symbols; SAS: blue × symbols.

environment. For instance, the turbulence intensity RMS at 8D was 0.03 in the low turbulence case, while it is 0.01 in the present case at 9D (measurements at the same location are not available in the two experimental data sets). A good estimation of turbulence intensity is necessary for the correct estimation of wake deficit. The good match observed here at 9D is therefore encouraging for the use of the present simulation models both for closely spaced wind farms, where the wake might be interacting with multiple machines, and for larger spacings, where one needs to account for impingement of wakes shed by machines far upstream.

In the near wake, a proper estimation of the effects of tip vortices can be observed, differently from the low turbulence case discussed in §5.1. The two turbulence intensity peaks can be clearly observed from 1.4D to 4D, covering the whole near wake range. It is possible that even a coarser grid could be used in this case, although a precise characterization of the degradation of the results with decreasing mesh density was not performed.

### 5.3.2 Single-turbine high-turbulence case

Next, a high turbulence (20%) condition is considered. In this case, experimental measurements are not available, and SAS is compared directly to LES, which serves as benchmark. The turbulent inflow is generated as previously explained, using the corresponding high-turbulence spires and wind tunnel configuration. The CPU time consumption ratio is $\mathrm{CPU_{LES}/CPU_{SAS}}$=9.37.




Results indicate a good agreement between SAS and LES, both in terms of velocity and of turbulence intensity, as shown in Fig. 7. Here again, one can notice that the difference between the two models tends to decrease in higher turbulence conditions. This is particularly true in the near hub region, which is probably due to the higher mixture created by the background turbulence.

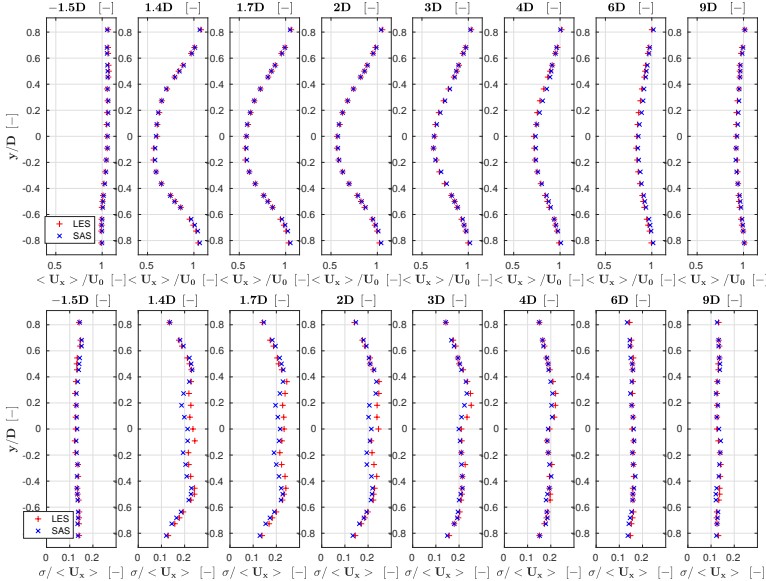

**Figure 7.** Normalized time-averaged streamwise velocity and turbulence intensity profiles at hub height at downstream positions: -1.5D, 1.4D, 1.7D, 2D, 8D, 3D, 4D, 6D and 9D. LES: red $+$ symbols; SAS: blue $x$ symbols.

### 5.4 Three aligned turbines

Results shown up to here indicate that SAS achieves in general a good agreement with LES both in terms of wake deficit and turbulence intensity. The match is of a better quality for increasing turbulence, while it is less satisfactory for low turbulence conditions. However, the moderate and high turbulence flows considered here represent more realistic atmospheric boundary layers, while very low turbulence conditions are less likely to be encountered in actual conditions in the field.

To characterize the performance of SAS in conditions characterized by wake interactions, three fully-waked wind turbines are considered, The machines are aligned with the flow, with their rotors pointing into the incoming wind and spaced among themselves of 4.1D. Experimental measurements are not available in this case.

The same moderate turbulence flow of §5.3.1 is used even in this case. The first upstream machine operates in the same exact conditions of the isolated wind turbine case, while the two downstream




machines are operated in closed by a pitch-torque controller, in order to adjust their operating point to the local incoming wind.

Figure 8 shows velocity and turbulence intensity profiles at various distances from the rotor. Notice that the wind turbines are located at 0D, 4.1D and 8.2D. The two simulation models show nearly identical velocity profiles at all downstream positions. The average error $\langle\Delta(u_x)\rangle$ over all stations is less than 1%. The turbulence intensity profiles are also in a good agreement, with only small discrepancies at 11D.

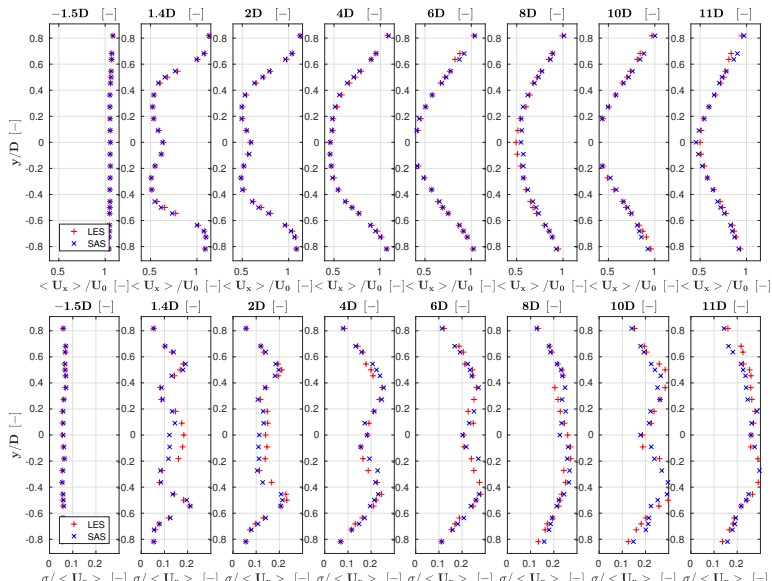

**Figure 8.** Normalized time-averaged stream-wise velocity (top) and turbulence intensity (bottom) profiles at hub height and at several positions, for a cluster of three waked turbines (located at 0D, 4.1D and 8.2D). LES: red + symbols; SAS: blue × symbols.

Figure 9 shows color plots of the normalized instantaneous streamwise velocity $u_x/U_0$, time-averaged streamwise velocity $\langle u_x\rangle/U_0$, vorticity magnitude $|\nabla\times\mathbf{u}|$ and turbulence intensity $\sigma/\langle u_x\rangle$ for LES and SAS on a vertical plane across the hub. Coherently with Fig. 8, the two turbulence models exhibit a nearly identical behavior for average velocity and turbulence intensity. The wake deficit behind the first wind turbine is accurately simulated by the SAS model, as already shown for the single turbine case. A correct modeling of the wake deficit behind the second and third wind turbine is more challenging, as wakes overlap and interact. In particular, the turbulence shed by the rotor enhances mixing between the wake and the free-stream flow. This effect has to be precisely resolved by the model for the correct wake deficit to be captured. For multiple interacting turbines in a row, as in the present case, errors or lack of accuracy in the modeling of these processes will





propagate and amplify dowstream, eventually increasing the simulation uncertainty at the back of
the cluster. However, velocity contours show a consistent behavior between LES and SAS even for
the second and third wake.

A more pronounced difference between the two models is apparent for the vorticity contour plots.
In fact, because of its higher mesh density, LES is capable of resolving finer vortical structures. This
is also noticeable for the blade root vortices, which are significantly diffused for SAS. Nonetheless,
these differences still appear to have only a negligible impact on the overall behavior of the wake.

## 6    Conclusions

In this paper, a scale-adaptive CFD formulation has been applied to the simulation of isolated and
wake-interacting wind turbines within different turbulent environments. The rationale for the use of
SAS is a desire to achieve an accuracy comparable to LES, but at a much reduced computational
cost.

The paper has first briefly reviewed the SAS formulation, as presented in the existing literature.
Then, an isolated wind turbine has been considered in a low turbulence inflow, to serve as baseline
test case for the tuning of the simulation environment. Next, other test cases have been studied,
considering inflows of increasing turbulence, cases where one machine is misaligned with the respect
to the main flow direction, and a case with three aligned and fully waked wind turbines.

From the results obtained in this work, the following conclusions may be drawn:

- In general, SAS appears to achieve a good agreement with both LES and experimental mea-
  surements in terms of time-averaged rotor integral quantities and velocity profiles. Results are
  not only qualitatively very similar, but also quantitative differences are typically limited to a
  few per cent points.

- In low turbulence conditions, blade tip-vortices appear not to be properly resolved by SAS. It
  was verified that this difference is not caused by the different modeling of turbulence between
  the two approaches, but it is in fact driven by the coarser mesh used by SAS. Because of the
  smearing of tip vortices, there is a defect of turbulence intensity in the near wake region for
  SAS, which in turn causes some small differences in far wake recovery. Although this problem
  is quite apparent in the low turbulence inflow conditions, it is less severe in higher ambient
  turbulence cases.

- LES required on average roughly 8 times larger meshes and 13 times longer CPU time than
  SAS.

As a general remark, one should keep in mind that moving from LES to SAS does not only entail
a different turbulence model but also implies the choice of less denser grids. While this is exactly
the reason that might motivate the use of SAS, one should remember that coarser grids will also





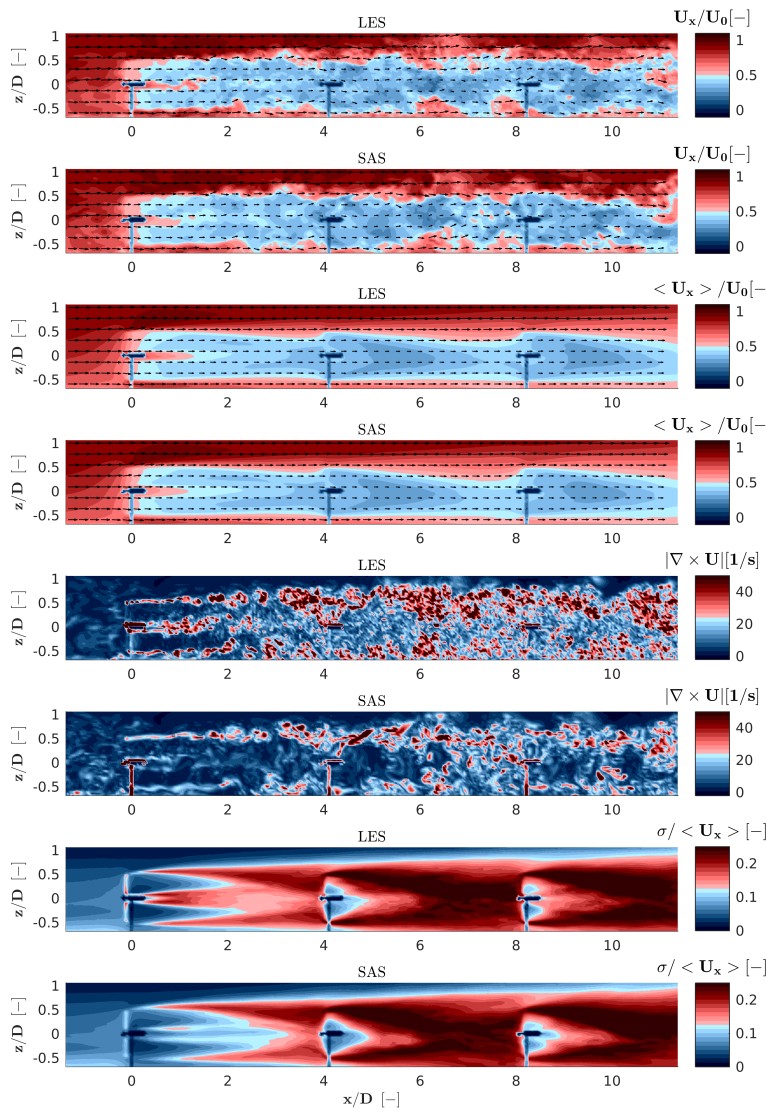

**Figure 9.** From top to bottom, normalized instantaneous streamwise velocity $u_x/U_0$, 60 s time-averaged streamwise velocity $\langle u_x \rangle/U_0$, vorticity magnitude $\|\nabla \times \mathbf{u}\|$ and turbulence intensity $\sigma/\langle u_x \rangle$, for both LES and SAS along a streamwise vertical plane across the hub.





imply effects that are not directly related to turbulence, as for example the accuracy of the lifting line calculations (both in terms of angle of attack computation and projection of the aerodynamic

525 forces back onto the fluid), the resolution of the flow around nacelle and tower, and the numerical diffusion of tip vortices.

Given that differences in results appear to be quite limited, at least in the conditions examined here, the significant difference in computational burden between the two methods indicates that SAS could be an interesting alternative to LES, at least in turbulent conditions. In this sense, one could

530 use SAS for tuning of wind plant control laws, repetitive runs or other tasks that do not require extremely accurate results, leaving LES for the final higher-accuracy runs.

*Acknowledgements.* This work has been supported in part by the CL-WINDCON project, which receives funding from the European Union Horizon 2020 research and innovation program under grant agreement No. 727477. The first author was supported by the Chinese Scholarship Council. All tests were performed

535 at the wind tunnel of the Politecnico di Milano, with the support of Prof. A. Croce, Mr. G. Campanardi and Mr. D. Grassi. The authors wish to thank Dr. V. Petrović, now at the University of Oldenburg, for his contribution to the experimental work, and Mr. Y. Liu for his support with the numerical simulations. The authors also express their appreciation to the Leibniz Supercomputing Centre (LRZ) for providing access and computing time on the SuperMUC Petascale System.

540 **Nomenclature**

| | |
|---|---|
| $c_\mu \ \sigma_k$ | Closure coefficients for turbulent kinetic energy transport equation |
| $k$ | Turbulent kinetic energy |
| $p$ | Pressure |
| $t$ | Time |
| **u** | Velocity vector |
| $u_x^{i,j}$ | Velocity component $x$ at time step $i$ and point location $j$ |
| $y^+$ | Dimensionless wall distance |
| $C_s$ | Smagorinsky constant |
| $D$ | Rotor diameter |
| $F_1$ | SST blending function |
| $F_\mathrm{SAS}$ | Scaling parameter for SAS model |
| $L$ | Length scale of the modeled turbulence |
| $L_{vK}$ | von Kármán length scale |
| $P_k$ | Production term |
| $Q_\mathrm{SAS}$ | SAS source term |
| $T$ | Temperature |
| $U_0$ | Free-stream flow speed |





|  | $\alpha, \beta, \sigma_\omega, \sigma_{\omega_2}$ | Closure coefficients for dissipation-rate transport equation |
|  | $\epsilon$ | Gaussian width |
| 560 | $\kappa$ | von Kármán constant |
|  | $\omega$ | Specific dissipation rate |
|  | $\nu$ | Molecular kinematic viscosity |
|  | $\nu_t$ | Kinematic eddy viscosity |
|  | $\rho$ | Flow density |
| 565 | $\sigma/\langle u_x \rangle$ | Turbulence intensity |
|  | $\sigma$ | Standard deviation |
|  | $\zeta_2, \sigma_\Phi, C, C_k$ | SAS model parameters |
|  | $\Omega_{CV}$ | Control volume |
|  | $\Delta \cdot$ | Difference between two quantities |
| 570 | $\langle \cdot \rangle$ | Averaged (in space and/or time) quantity |
|  | $\tilde{(\cdot)}$ | Resolved quantity |
|  | $(\cdot)_x$ | Streamwise component |
|  | $(\cdot)_y$ | Lateral component |
|  | $(\cdot)_z$ | Vertical component |
| 575 | ALM | Actuator line method |
|  | CFD | Computational fluid dynamics |
|  | IB | Immersed boundary |
|  | LES | Large-eddy simulation |
|  | LiDAR | Light detection and ranging |
| 580 | LU | Lower and upper matrix |
|  | PISO | Pressure implicit with splitting of operator |
|  | RMS | Root mean square |
|  | SAS | Scale-adaptive simulation |
|  | SST | Shear stress transport |
| 585 | URANS | Unsteady Reynolds-averaged Navier Stokes |



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
