# Peer review of "Scale-adaptive simulation of wind turbines, and its verification with respect to wind tunnel measurements"

_Wind Energy Science, 2018_

## Referee Comment (RC1) · Anonymous Referee #1 · 24 Sep 2018

Dear authors,

thank you very much for the interesting paper. As a reviewer I have quite some remarks, although most of them are not so severe. On the other hand some of the questions I have are really so severe, that I do insist on having them clarified before publishing:

1. In the literature in the introduction: Of course there are also models in between quick "analytical" models, like the wake meandering model and LES. E. g. there also has been quite some research on RANS simulations.

2. Also there has been a lot more research done on LES of wakes with a lot of different codes (like in Leuven, DTU or Oldenburg).

3. I believe it is "an LES code" not "a LES code", because you pronounce the "L" like "Al".

4. You site papers which are still under review. This is of course hard to judge, since it is unclear, if these papers are scientifically profound. I don't now how to deal with it. Are they accepted yet?

5. Please also discuss critical points on the SAS model: Is it able to capture anisotropic flows? How about rotating flows?

6. In line 120: From which experiments were the parameters taken? What were the values you used?

7. In equation 3 please define all parameters also specify, if Ck is an SAS parameter, which value you used there.

8. In line 151, what is a complete digital copy? Is that possible? This would mean to also copy all boundary conditions exactly (roughnesses etc.). Also a question would be: Is the an influence by the wind turbine on the incoming flow without turbine – or this way: Has the induction of the turbine an influence on the incoming flow and its turbulence? If so, then the assumption to use the same turbulent inflow for different simulations is an approximation.

9. In lines 165-166: In which boundary layer you used a y+ of 50? Is it on the wind tunnel walls? Is it on the spires?

10. In general, the paper does not give a good view on the setup. What does the wind tunnel look like? How did you mesh it? If this has been described in a different paper, which is also still under review and might never be published, how shall I deal with this? I would propose to include more of the description of tunnel and the mesh also of the precursor run into this paper.

11. In lines 177-180 you mention, that the low turbulence cases are simulated from data coming from a LiDAR measurement by prescribing a time constant velocity. This

is a bit unclear to me: The LiDAR is a scanning device. It gives you different velocities (or rather components of the velocities) at different points in time and space. You did you prescribe a time constant velocity there?

12. In line 187 you mention, that your 3.6D domain would avoid blockage effects. Now I know a lot of simulations with a wider domain. How can you be sure, you don't have any blockage effects?

13. While the SAS model has been discussed a bit, the LES model remains rather unclear: Which LES-SGS model did you use? Which parameters did you use in it?

14. For the LES case: Did you do a grid refinement study? Maybe a coarser mesh in the LES case would also give you good results? How do you know, this is not the case?

15. The sentence line 203-205 is somehow strange . . .

16. Did you test iLES – so without SGS-model?

17. In 3.2.2 how did you use T? Did you take thermal effects into account? Was T non homogeneous and why?

18. If you change the boundary conditions on the tower due to turbulence: What is the effect on the results if you turn from non-slip to slip?

19. In 3.2.4 you specify the LES model, however this show be discussed with the turbulence modelling together with the SAS model,

20. Why did you put Cs=0.13?

21. Why did you use 2.5 for epsilon in the AL?

22. Why did you use Fsas = 2?

23. paragraph from line 268 to 274: Now, if the wake is not much influenced by the loads, is it much of a surprise, that different turbulence models (in the LES framework)

do not show much difference?

24. In chapter 5 you also do compare to k-omega SST (RANS), now, what was the grid there? What was the computational speed? What was the time stepping?

25. The power output predicted by the SST model ist only a little bit worse than the one of the SAS model. So what is the advantage using SAS in terms of power prediction?

26. In figure 2 – well this looks very strange! It looks like you used a 3 bladed turbine for LES and a one bladed for SAS and k-omega SST? Is this so? How did you compare then? Is this a fair comparison? Why?

27. You calculated RMSE and TI for certain distances from the turbine. How broad was the field you evaluated for this? 28. The part on the symmetry after line 350 is unclear. What is the reason for this phenomenon?

29. In some parts you compare SAS to LES. However the wording sometimes is like "SAS overestimates ..:". This assumes, that LES is correct. How do you know? Obviously LES also has some deviations (sometimes even different deviations than SAS).

30. In line 360 on the k-omega SST model (is it RANS or DES k-omega SST??), well, you did not specify the grid resolution, so it is completely unclear why the model results behave in such a way. From figure 2 I would guess, you used only one blade on a far to coarse mesh for the wake. Other simulations using the same model showed better wake properties.

31. From line 392 on you use the terms "left" and "right" for descriptions of the wakes. This is not a good way to describe, since it always needs a definition of left and right from which viewpoint, with a turbine turning in which way? Maybe you can find a more descriptive way of defining the sides?

32. In lines 407 following, what does it show, if you leave away the nacelle and tower? What does this proof? You are a bit unclear there.

33. In the cases of the turbulent inflow: How much does the result depend on the integral length scale of the turbulence? What was the integral length scale – also in comparison to grid size? What was it in the wind tunnel experiment?

34. It would be very helpful if you analyze the power density spectrum of the turbulence once (experiment and LES). How much energy do you loose, if you use a coarser grid? How much is captures by the actual SAS model?

---

## Referee Comment (RC2) · Anonymous Referee #2 · 3 Oct 2018

article

**Summary**

The article describes the potential of using an scale-adaptive URANS model based on the k-omega-SST model as an alternative to LES for the simulation of wind turbine wakes.

The article is well written and organized. Furthermore, the topic is of sufficient scientific

relevance for the wind energy community. However, it is imperative that some important issued are addressed before this referee can recommend it for publication. My major objections are related to the lack of a grid convergence study. This makes nearly impossible to compare and interpret the results with any confidence. Consequently the conclusions of this work lack of a solid foundation.

In the following, I describe both major and minor issues that should be adressed.

**Major comments:**

1. My major concern is the lack of a grid convergence study. How do you know that the mesh resolution is enough for LES? How do you know that it could not be coarser? What about the same questions for the URANS simulations? In the way you present your results, this referee can not have any confidence that your interpretation of the results is correct.

2. Because of the lack of a grid convergence study, I wonder if LES with the RANS grid would provide similar results to $k - \omega - SST - SAS$. Perhaps it produces similar results at a lower computational cost, since it needs to resolve less equations. Please clarify first the issue with the grid uncertainty, and then you will be in the position to discuss the benefits and drawbacks of the different turbulence modelling strategies. I suggest the use of Roache's Grid Convergence Index (GCI).

3. You use the Gaussian smearing factor $\epsilon = 2.5$, but you do not say why. In the literature, it is usually recommended to choose $\epsilon = 2.0$ (see e.g. PhD thesis by Troldborg from DTU). For lower $\epsilon$, the simulation usually tends to be unstable, and for larger $\epsilon$ it tends to be inaccurate. In fact, the correct tuning and setting of this parameter always requires a sensitivity analysis. Please include it.

[Figure]

4. Additional details on the actuator line model are required: do you use a tip correction model? If yes, which and why? What about root corrections? Which is the number of elements per blade? How is the spacing between the elements? Do you use dynamic-stall corrections? Which is the ratio between the average chord length and the cell size?

5. The $k - \omega - SST - SAS$ are partly more accurate than the LES - Smagorinsky results (e.g. line 343). How can this be? Please explain this unexpected behaviour.

6. line 384: For LES the tip chord length it 1.8 times the cell size. For RANS it is then 3.6 times the cell size? According to the recommendations from the literature, the average chord length should be around the same size as the cell size! And you are talking about the tip region, so for the rest of the blade this issue is even more pronounced! Please elaborate on this issue.

7. line 396: I am not sure about the wake behaviour you describe here. In case the rotor and tower interact in the way here proposed, why does it just move slightly upward? Actually the part of the tower shadow comprising one rotor lenght should rotate in the same way that the rotor wake does. And the same effect should be also visible in the case without yaw misalignment. How do you distinguish the tower wake from the rotor wake after they meet?

8. A discussion of the experimental uncertainty is required.

**Minor comments**

1. The nomenclature that you use for the different turbulence modelling approaches is quite misleading. What you are comparing is LES with a Smagorinsky subgridscale model, URANS with $k - \omega - SST - SAS$, and URANS with $k - \omega - SST$. Please adapt the whole text accordingly.

2. The scale adaptive model SAS is an URANS model and is based on k-Omega-SST. This must be explicitly described in the paper.

3. Please explain all the parameters of all the equations.

4. Page 4, line 34: The available blade conforming approaches do NOT include LES because of its huge computational cost. The article that the authors are citing is based on DES not LES.

5. All the turbulence models that you use are readily available in OpenFOAM. Please state this information, so that other researchers can reproduce your work.

6. page 7, lines 196-198: You mention the general results of your work several times before and after this point. In this section you are supposed just the numerical mesh. Please delete this sentence and try to avoid describing the results before coming to the results section.

7. From the beginning of the article, it was clear which turbulence models you use for RANS, however it was unclear which sub-grid scale model you use for LES. I found that information in section 3.2.2 when you describe the boundary conditions, but up to that point I was wondering the whole time about it. Please state this information more clearly at a earlier point.

8. If I understand it right, you do NOT use the *dynamic* Smagorinsky model for LES. I wonder why.

9. Line 238: you use the van Leer differencing scheme and you claim that it is strictly bounded. However, I believe that the implementation in OpenFOAM is unbounded. Please clarify this.

10. Line 256: it is unclear why you assume $F_{SAS} = 2$

11. Section 4: why do not you use the same controller for the experiments and the simulations?

12. Line 358: you claim that SAS may lead to a faster wake recovery. Why "may"? Does it or does it not?

13. Fig. 5: Are the results time-dependant or averaged?

14. line 424: what about the thrust?

15. line 484: I think that you do not mean "shed" but "trailed". Please correct or comment on this.